# Twisted Few-Mode Optical Fiber with Improved Height of Quasi-Step Refractive Index Profile

**DOI:** 10.3390/s22093124

**Published:** 2022-04-19

**Authors:** Anton V. Bourdine, Vladimir V. Demidov, Artem A. Kuznetsov, Alexander A. Vasilets, Egishe V. Ter-Nersesyants, Alexander V. Khokhlov, Alexandra S. Matrosova, Grigori A. Pchelkin, Michael V. Dashkov, Elena S. Zaitseva, Azat R. Gizatulin, Ivan K. Meshkov, Airat Zh. Sakhabutdinov, Eugeniy V. Dmitriev, Oleg G. Morozov, Vladimir A. Burdin, Konstantin V. Dukelskii, Yaseera Ismail, Francesco Petruccione, Ghanshyam Singh, Manish Tiwari, Juan Yin

**Affiliations:** 1Department of Communication Lines, Povolzhskiy State University of Telecommunications and Informatics, 23, Lev Tolstoy Street, Samara 443010, Russia; bourdine@yandex.ru (A.V.B.); mvd.srttc@gmail.com (M.V.D.); zaytzewa@inbox.ru (E.S.Z.); burdin@psati.ru (V.A.B.); 2JSC “Scientific Production Association State Optical Institute Named after Vavilov S.I.”, 36/1, Babushkin Street, St. Petersburg 192171, Russia; demidov@goi.ru (V.V.D.); ter@goi.ru (E.V.T.-N.); khokhlov@goi.ru (A.V.K.); a.pasishnik@gmail.com (A.S.M.); beegrig@mail.ru (G.A.P.); kdukel@mail.ru (K.V.D.); 3“OptoFiber Lab” LLC, Skolkovo Innovation Center, 7, Nobel Street, Moscow 143026, Russia; 4Department of Photonics and Communication Links, Saint Petersburg State University of Telecommunications Named after M.A. Bonch-Bruevich, 22, Bolshevikov Avenue, St. Petersburg 193232, Russia; 5Department of Radiophotonics and Microwave Technologies, Kazan National Research Technical University Named after A.N. Tupolev-KAI, 10, Karl Marx Street, Kazan 420111, Russia; a.vasilets@mail.ru (A.A.V.); azhsakhabutdinov@kai.ru (A.Z.S.); microoil@mail.ru (O.G.M.); 6Faculty of Management and Engineering Business, Kazan Innovative University Named after V.G. Timiryasov (IEML), 42, Moskovskaya Street, Kazan 420111, Russia; 7Faculty of Photonics and Optical Information, School of Photonics, ITMO University, Bldg. A, 49, Kronverksky Alley, St. Petersburg 197101, Russia; 8Institute of Physics, Nanotechnology and Telecommunications, Peter the Great St. Petersburg Polytechnic University, Bldg. II, 29, Politekhnicheskaya Street, St. Petersburg 194064, Russia; 9Department of Telecommunication Systems, Ufa State Aviation Technical University, 12, Karl Marx Street, Ufa 450000, Russia; azat_poincare@mail.ru (A.R.G.); mik.ivan@bk.ru (I.K.M.); 10Department of Quantum Communication Systems Research, Radio Research and Development Institute, 16, Kazakova Street, Moscow 105064, Russia; dmitriev@niir.ru; 11Quantum Research Group, School of Chemistry and Physics, University of KwaZulu-Natal, Durban 4001, South Africa; ismaily@ukzn.ac.za (Y.I.); petruccione@ukzn.ac.za (F.P.); 12National Institute for Theoretical Physics, University of KwaZulu-Natal, Durban 4001, South Africa; 13Department of Electronics and Communication Engineering, Malaviya National Institute of Technology (MNIT) Jaipur, J.L.N Road, Jaipur 302017, Rajasthan, India; gsingh.ece@mnit.ac.in; 14Department of Electronics and Communication Engineering, School of Electrical and Electronics & Commu-nication Engineering, Manipal University Jaipur, Jaipur-Ajmer Expressway, Jaipur 303007, Rajasthan, India; manish.tiwari@jaipur.manipal.edu; 15Division of Quantum Physics and Quantum Information, University of Science and Technology of China, 99, Xiupu Road, Pudong District, Shanghai 200093, China; yinjuan@ustc.edu.cn

**Keywords:** twisted optical fiber, laser beam profile, differential mode delay, laser-based few-mode optical signal transmission, fiber Bragg grating, few-mode effects

## Abstract

This work presents designed and fabricated silica few-mode optical fiber (FMF) with induced twisting 10 and 66 revolutions per meter, core diameter 11 µm, typical “telecommunication” cladding diameter 125 µm, improved height of quasi-step refractive index profile and numerical aperture 0.22. Proposed FMF supports 4 guided modes over “C”-band. We discussed selection of specified optical fiber parameters to provide desired limited mode number over mentioned wavelength range. Some results of tests, performed with pilot samples of manufactured FMF, are represented, including experimentally measured spectral responses of laser-excited optical signals, that comprise researches and analysis of few-mode effects, occurring after fiber Bragg grating writing.

## 1. Introduction

Twisted optical fibers have been known since the early 1980s: here, the concept of fiber spinning was firstly originally introduced in the work [1]. The fabrication technique of twisted optical fibers is based on rotation of preform during the fiber drawing [1] or directly spinning of the drawn optical fiber [2]. Twisted single mode optical fibers are usually considered as fibers with reduced polarization mode dispersion (PMD) [1,2,3,4], while induced chirality (including twisting) over multimode optical fibers is declared as the method for differential mode delay (DMD), decreasing with total bandwidth improvement [5,6].

Nowadays, twisted optical fibers (both with typical coaxial geometry (core, bounded by intermediate and/or outer solid cladding) and microstructured/photonic crystal optical fibers) are considered as alternative unique fiber optic elements with great potentiality for various applications in fiber optic sensors [7,8,9,10].

At the same time, many recently published works demonstrated new effects, occurring in fiber Bragg gratings (FBGs), written in FMFs as well as in multimode optical fibers (MMFs), under laser-source excitation. A lot of many recently published works demonstrated few-mode operation of these conventional, tilted or slanted FBGs on MMFs and FMFs in vibration, temperature, deformation, displacement, bending, etc., fiber optic sensors [11,12,13,14,15,16,17,18,19,20].

A few-mode regime adds a new other dimension to the space of parameters: it is associated with guided modes of particular order, in which a limited number (from two to a few dozen) transfer the most part of optical signal power over tested optical fiber. We suppose that twisted FMF with recorded FBG can be considered as a new complicated fiber optic element with unique features and great potentiality for application in fiber optic sensors.

This work is focused on design and fabrication of twisted few-mode optical fiber (FMF) with specified limited (3…4) guided modes, supporting over “C”-band. Therefore, at the first stage, by using commercially available software with rigorous numerical finite element method, technological/geometrical parameters were specified to provide the desired few-mode regime operation of designed FMF. 

We performed adaptation of the conventional technique for drawing optical fibers to fabricate designed FMF with induced twisting under small as well as large number of revolutions per meter. Some results of theoretical and experimental researches, performed for pilot samples of manufactured FMF, are represented, including experimentally measured spectral responses of laser-excited optical signals, that comprise researches and analysis of few-mode effects, occurring after fiber Bragg grating recording.

## 2. Design of FMF: Selection of Geometrical Parameters to Provide Desired Limited Number of Guided Modes

At the first stage, we utilized rigorous numerical finite element method, used in COMSOL Multiphysics software, to analyze a preliminary designed set of specified step-index optical fibers with the same typical “telecommunication” cladding diameter 125 µm, but differing by combination of core diameter and numerical aperture (e.g., height of refractive index profile). Here, for each designed optical fiber sample, guided modes (which satisfy to the cut-off condition) were defined, and their effective refractive indexes were computed. The main criterion was focused on providing limited (3…4) transversal guided modes′ propagation over FMF under laser source excitation at the wavelength λ = 1550 nm.

We performed analysis of FMF by an earlier developed and successfully verified method [21,22,23], based on the numerical solution of linear Maxwell equation system, written for a homogeneous isotropic dielectric in the absence of free charges and currents and reduced to wave equations for the vectors of electric (*E*) and magnetic (B) fields [24]:(1)$$\nabla \times \left(\frac{1}{\mu }\nabla \times E\right)-k_{0}^{2}\varepsilon E=0,$$
where *k*_0_ is wave number (*k*_0_ = 2π/λ); ε is the dielectric permeability (ε = *n*^2^, *n* is refractive index); μ is magnetic permeability.

By taking into account satisfaction to the perfectly matched layer (PML) conditions, Equation (1) was transformed to the following form [25]:(2)$$\nabla \times \left(\frac{1}{\mu \prime }\nabla \times \frac{1}{\left[S\right]}E\right)-k_{0}^{2}\varepsilon^{\prime }\frac{1}{\left[S\right]}E=0,$$
where ε′ and μ′ are modified dielectric and magnetic permeability; [*S*] is matrix of PML layer coefficients.

Solution of Equation (2) is equation of electromagnetic wave, propagating along *z*-axis of optical fiber [24]:(3)$$E\left(z,\;t\right)=E_{0}\exp\left[j\left(\omega t-\frac{\omega }{c}n_{\mathrm{eff}}z\right)\right],$$
where *E*_0_ is amplitude of electric field strength; ω is circular frequency; *c* is light speed in vacuum; *t* is time.

Effective refractive index *n*_eff_ is defined by numerical solution of Equation (3) and related transversal mode is identified (in terms of linear polarized modes *LP_lm_*, e.g., azimuthal and radial orders *l* and *m*) by comparison computed radial mode field distribution with pre-defined field patterns of known order modes *LP_lm_*, which are exact solutions for scalar wave equation, written for model optical fiber with ideal step index or unbounded parabolic refractive index profiles and described by Bessel or Laguerre-Gaussian functions [26,27].

Table 1 shows results of optical fiber analysis, performed by a rigorous finite element numerical method in COMSOL Multiphysics software under wavelength λ = 1550 nm. We considered ideal step-index refractive index profile, the same cladding diameter 125 μm (that corresponds to conventional telecommunication optical fibers), three various core diameters (8.3, 10.0 and 11.0 μm) and six values of numerical aperture *NA* (0.14, 0.16, 0.18, 0.20, 0.22 and 0.24—it corresponds to approximately the difference between core and cladding refractive indexes 0.02). We start from the core diameter 8.3 μm as the typical value for standard single mode optical fibers (SMFs) of ITU-T Rec. G.652 [28]. It was supposed that even the weak improvement of refractive index profile height, in comparison with ratified SMF, may provide desired few-mode regime with 3…4 transversal guided mode propagation at λ = 1550 nm. However, results of computation showed that following increasing both core diameter and numerical aperture (e.g., refractive index profile height) is required). 

For example, combination of the SMF “nominal” core diameter 8.3 μm and maximal (from the researched range) numerical aperture value *NA* = 0.24 provides technical satisfaction of the cut-off condition for desired 4 modes—the fundamental *LP*_01_ and higher order modes *LP*_02_, *LP*_11_, *LP*_21_. However, the last two modes *LP*_02_ and *LP*_21_ are unacceptably instable to propagation over long distances due to their field concentration in the cladding: here, optical confinement factor *P_co_* (e.g., mode power, transferred over core) for both aforementioned modes is inadmissibly low (*P_co_* < 0.5). Therefore, we conclude that none of the researched combinations of core diameter 8.3 μm and 6 tested numerical aperture values *NA* = 0.14…0.24 do not provide desired 4-mode operation at wavelength λ = 1550 nm. The same matter corresponds to core diameter 10.0 μm and *NA* = 0.20: here also, higher-order modes *LP*_02_ and *LP*_21_ satisfy cut-off condition under unacceptable low optical confinement factor *P_co_* < 0.5, while desired 4-mode operation is achieved for numerical aperture range *NA* = 0.22…0.24. Following improvement of core diameter up to 11 μm showed the best results for *NA* = 0.20 and *NA* = 0.22: all 4 modes satisfy the cut-off condition under the required optical confinement factor *P_co_* > 0.5. Lower *NA* = 0.18 led to inappropriate low *P_co_* < 0.5 for the same last two higher-order modes, while increased *NA* = 0.24 provides satisfaction of the cut-off condition for 5th mode *LP*_31_.

Therefore, according to computation results, we selected the following configuration for fabricated FMF: core diameter 11 µm, typical “telecommunication” cladding diameter 125 µm, numerical aperture *NA* = 0.22.

## 3. Pilot FMF 11/125 with Improved Height of Quasi-Step Refractive Index Profile and Induced Twisting

According to the aforementioned technological parameters, a preform of the desired FMF 11/125 with the numerical aperture *NA* = 0.22 was prepared by conventional modified chemical vapor deposition (MCVD) method [29]. Figure 1 presents measured refractive index profile with improved height of FMF fabricated preform. 

The general form of fabricated preform refractive index profile is quasi-step. Moreover, there is a dip of refractive index in the core center, which is typical for MCVD technique: it is caused by highly volatile GeO_2_ dopant diffusion during support tube collapse. Here, the absolute height of the profile reaches ~0.27, while dip is ~0.08. As a result, to correctly evaluate refractive index profile height, we computed the area of the central (core) part and further estimated the effective height of the profile. For researched prepared FMF 11/125 preform, this parameter was ~0.018, that is equivalent to numerical aperture *NA* = 0.22.

We performed some modifications of the drawing tower to induce twisting on FMF during its drawing. The detailed description of modification is represented in the earlier published work [30]. Usually, preform is fixed in a mechanical chuck of the feed unit, which inputs preform to the heat space of a high temperature furnace. Preform is kept in a stationary position and redrawn without rotation. To induce desired twisting over manufactured FMF, we integrated the stepper motor to the feed unit, which continuously rotates preform under the set speed and adds a new rotation function to the drawing system. The minimal motor rotation speed is 20 revolutions per minute, while the maximal is 200. Therefore, under slow drawing speed 2…3 m/min (that is usually used for manufacturing special or experimental optical fibers), it induces twisting with 10 and 66 revolutions per meter (rpm), respectively.

Figure 2 shows an image of the end-face of fabricated pilot sample of twisted FMF 11/125 with numerical aperture *NA* = 0.22, drawn from the aforementioned manufactured preform. Figure 3 presents near field laser beam profile (operating wavelength λ = 1550 nm) after propagation over the fabricated FMF 11/125 by CCD camera.

We measured both 10 rpm and 66 rpm pilot sample 50 m length FMF 11/125 attenuation α(λ) by cutback method over wavelength band λ = 900–1700 nm by using a halogen lamp (OSRAM 64642 HLX) as a light source, programmable monochromator (ANDO), germanium photodiode (wavelength range 900…1700 nm), optical amplifier (eLockIn) and optical power meter (and ANDO AQ-1135E). 

Measured attenuation curves α(λ) contain typical resonance “water” peaks with strong loss due to simplified and rapid technique for FMF preform fabrication without hydroxyl (OH^–^) dopants extraction: here, we just focused on twisted FMF pilot sample length manufacturing with specified geometry parameters, which should provide desired few-mode operation by low widening core diameter and strong improvement of refractive profile height and did not pay attention to attenuation reduction. In the same way, increased attenuation (in comparison with commercially available silica optical fibers [28]) at the central regions of the “C”- and “O”-bands, which reaches almost α = 7…8 dB/km, was expected due to intentional excluding (to reduce reagent consumption and also to simplify preform fabrication process) of typical operation of Fluorine (F) doping to the core region, which helps to decrease GeO_2_ dopant unwanted influence on attenuation increasing.

Figure 4 demonstrates that attenuation curve α(λ) over “flat” regions between the resonance peaks for FMF with twisting 66 rpm being lower in comparison with 10 rpm twisted FMF. It may be explained by more smoothing of refractive index profile typical MCVD technological defects under more rapid twisting of preform during optical fiber drawing.

## 4. Dispersion Parameters of Guided Modes, Propagating in the Pilot Sample FMF 11/125 with Improved Height of Quasi-Step Refractive Index Profile

During the next stage, we computed spectral characteristics of dispersion parameters of guided modes, satisfying the cut-off condition for fabricated pilot sample of FMF 11/125. For this purpose, it was proposed to utilize an earlier on developed simple and fast approximate method, which is a modification of the Gaussian approximation, extended to the case for estimation of the transmission parameters of arbitrary order modes, propagating in a weakly guiding optical fiber with an arbitrary axially symmetric refractive index profile [31], with following, optionally (in appropriate case), accuracy improvement by rigorous numerical method of mixed finite elements [32]. This extended modification of the Gaussian approximation (EMGA) is based on combination of the stratification method [26] and “classical” Gaussian approximation [27]. Stratification method provides ability to represent complicated form of researched optical fiber refractive index profile with high detailing and corresponding technological defects (including local refractive index fluctuations), in spite of the most approximate methods, which typically utilize one or a set of smooth functions. Proposed approach significantly reduces computational error during direct calculation of transmission parameters of guided modes in optical fiber with large core diameter (in comparison with single mode optical fibers) and complicated form of refractive index profile [31,32]. Here, only one variational parameter—normalized mode field radius *R*_0_—should be determined as a result of characteristic equation solution, while *R*_0_ within the “classical” Gaussian approximation is the basis and it completely defines all desired guided mode transmission parameters. According to Gaussian approximation, radial mode field distribution is represented by a well-known approximating expression, based on Laguerre-Gaussian functions [27], that corresponds to exact solution of scalar wave equation, written for weakly guiding optical fiber with an ideal inbounded parabolic refractive index profile. This permits to derive and write analytical expressions for variational expression and characteristic equation in the form of finite nested sums, and further, their first and second derivations—mode delay and chromatic dispersion parameter. Therefore, developed approximate method EMGA does not require high computational resources (even during higher-order mode dispersion parameter estimation) and provides low (less than 1% [26,27]) computational error.

Figure 5 shows an equivalent quasi-step refractive index profile of the analyzed FMF 11/125 with a numerical aperture *NA* = 0.22, restored by report of measurements, performed for drawn optical fiber.

At the first stage, we computed optical confinement factor for modes, propagating in a mentioned above FMF 11/125, over wavelength range λ = 700…1700 nm. The results are presented in the form of a diagram in Figure 6.

According to computational results, desired 4-mode optical signal transmission is provided by researched FMF 11/125 over band λ = 1450…1700 nm. Generally, 38 *LP_lm_* modes with *l* = 0…7 azimuthal and *m* = 1…9 radial orders nominally satisfy the cut-off condition at the least researched wavelength range bound λ = 700 nm. However, only for 19 modes with also *l* = 0…7, but *m* = 1…4, orders optical confinement factor as more *P_co_* ≥ 0.5 for the same wavelength.

At the central region of the “O”-band (λ = 1300 nm), researched FMF 11/125 supports 6 guided modes that satisfy the cut-off condition under the optical confinement factor value more *P_co_* ≥ 0.5: they are listed in Section 2—*LP*_01_, *LP*_11_, *LP*_21_, *LP*_02_ modes and two additional higher-order modes *LP*_12_, *LP*_31_. We computed spectral curves of dispersion parameters for those 6 aforementioned guided modes. Results are represented in Figure 7a with spectral characteristics of mode delay and Figure 7b with chromatic dispersion coefficient. Analysis of mode delay curves show that DMD reaches 18.35 ns/km over λ = 1300 nm wavelength region, while near λ = 1550 nm DMD decreases down to 14.93 ns/km due to “suppression” of two higher-order modes *LP*_12_ and *LP*_31_.

By comparing spectral characteristics of the chromatic dispersion coefficient for the fundamental and higher-order modes, computed curves are generally similar to spectral characteristic of chromatic dispersion coefficient for standard telecommunication single mode optical fiber (SMF) of ITU-T Rec. G.652 [28]. Here, zero dispersion wavelength of both the fundamental and higher-order guided modes corresponds to wavelength range λ = 1300…1350 nm. Maximal deviation of this parameter *D* between higher-order guided modes was 27.09 ps/(nm·km) at λ = 1300 nm and 4.97 ps/(nm·km) at λ = 1550 nm.

## 5. Experimental Research of Spectral Responses of FBG, Written over Twisted FMF 11/125 with Improved Height of Quasi-Step Refractive Index Profile

Two FBG samples were written on the short (less 1.5 m) segments of SMF (Rec. ITU-T G.652 [28]) and fabricated pilot sample of twisted FMF 11/125 by Lloyd ineterferometric setup workstation under the same mask (with the same grating period), providing expected Bragg wavelength about λ_B_ ≈ 1550 nm. We performed preliminary measurements of both FBG spectral responses under propagation of optical signal, generated by continuous emission (CE) wideband laser diode (LD) with operating wavelength λ = 1550 nm and pigtailed by SMFs. Conventional setup was utilized for FBG spectral response measurement by optical spectrum analyzer (OSA) with fiber optic circulator (CIR), also pigtailed by SMFs. The described above scheme for testing of FBG, written on FMF, is shown in Figure 8. Both tested FBGs were jointed to SMF pigtail by fusion splicer and further connected to circulator via corresponding fiber optic adapter.

Results of measurements—OSA software screenshots—are represented in Figure 9a,b. Comparison of two measured spectral responses show that detected Bragg wavelength of FBG, written on FMF 11/125, is higher up to 16.46 nm (λ_B_ = 1567.50 nm), than for FBG on SMF (λ_B_ = 1551.04 nm). This suggests that effective refractive index for the fundamental mode *LP*_01_ of FMF is somewhat higher, in comparison with the fundamental mode *LP*_01_ of SMF, in approximately 1%. Spectral response of FBG on FMF contains main and periphery peaks. It may be considered as superposition of several modes, corresponding to transversal mode components, that led to response widening and confirms desired few-mode regime of FMF operation at the central wavelength of “C”-band (λ = 1550 nm).

The next set of tests was concerned with research of FBG Bragg wavelength λ_B_ shifting sensitivity to the temperature action with the following comparison. We placed sequentially both FBGs to the thermostat and discretely varied temperature from +40 °C up to +120 °C with a step of 20 °C. Here, Bragg wavelength λ_B_ under the least bound temperature +40 °C was considered as the reference value for the following estimation of λ_B_ shifting under the temperature increasing. Results are represented in Figure 10. Both dependences are highly linear, while the slope for FBG on FMF is somewhat higher (approximately on 5%).

By analogy with the previous measurements, we performed test series which focused on research and comparison sensitivity of FBGs on SMF and FMF to mechanical action. For these researches, we placed FBGs to the precision translation stage, which provides tensile and following with precision particular elongation of researched optical fiber segment with written FBG over the range 100…250 µm with a step of 50 µm. Here, Bragg wavelength under the unstrained state was considered as the reference value for the following estimation λ_B_ shifting Δλ under the described mechanical action. Results are represented in Figure 11. Both dependences are also highly linear, while the slope for FBG on FMF is somewhat higher (approximately on 2.5%).

The next test series was concerned with researches of few-mode effects, occurring during laser-excited optical signal propagation over FBG, written in FMF 11/125, under some various stress actions. Here, we utilized a “direct” FBG spectral response measurement scheme without fiber optic circulator (Figure 12). We tested sample FBG on FMF which was pigtailed by using fusion splicer by short SMF pigtails with length not more than 140 mm to avoid conversion of higher-order FMF guided modes to leakage/cladding modes in SMF pigtail [33]. 

Spectral response, measured for FBG on FMF 11/125 in an unperturbed state (that would be further considered as the reference) is presented in Figure 13. Here, 3 peaks (1 main peak (1567.24 nm) and 2 periphery peaks (1566.65 nm and 1567.96)) could be seen quite distinctly. During the next tests, we measured spectral responses under forming FMF loop with radius 15 mm before, on and after FBG. Results are shown in Figure 14. As expected, in all cases, Bragg wavelength shifting was detected. However, response smoothing as well as periphery peak dropout were noticed. Here, λ_B_ shifted down to Δλ = 0.24 nm under loop before and after FBG, while the loop on FBG λ_B_ became longer up to 0.08 nm in comparison with the reference response main peak value. Second test series was concerned with spectral response measurements after placing loops with radius 86 and 63 mm over researched segment of FMF 11/125 with written FBG. Results are demonstrated in Figure 15. Here again, response smoothing and periphery peak dropout are noticed under the same Bragg wavelength shifting down to 0.16 and 0.20 nm.

## 6. Conclusions

This work is devoted to the design and fabrication, as well as experimental and theoretical researches of the parameters of FMF 11/125 with induced twisting and improved height of a quasi-step refractive index profile, which provides 4-mode operation over “C”-band. Based on the series of simulation of described optical fiber, we selected specified technological parameters to support the desired 4 guided modes over the mentioned above “C”-band: core diameter 11 μm, cladding diameter 125 μm and numerical aperture *NA* = 0.22.

Successfully fabricated pilot sample lengths of the described above FMF 11/125 with induced twisting of 10 and 66 rpm are presented. Results of measured attenuation showed expected increased loss α = 7…8 dB/km over “C”- and “O”-bands, explained by an intentionally simplified technique for FMF preform manufacturing by excluding typical F doping to optical fiber preform core region, which helps to decrease GeO_2_ dopant unwanted influence on attenuation increasing. 

We performed analysis of designed and fabricated FMF 11/125: here, data from the measurement report were utilized to restore the real form of quasi-step refractive index profile. Orders of guided modes, satisfying the cut-off condition, were defined over researched wavelength band λ = 700…1700 nm (4…6 guided modes were localized over “C”- and “O”-band, respectively). Spectral characteristics of dispersion parameters (mode delay and chromatic dispersion coefficient) for defined guided modes were computed for the mentioned above researched wavelength range. Analysis of the mode delay curves showed that at the wavelength λ = 1300 nm, DMD reaches 18.35 ns/km, while at the wavelength λ = 1550 nm, it reduces down to 14.93 ns/km. By comparing spectral characteristics of the chromatic dispersion coefficient for the fundamental and higher-order modes, computed curves are generally similar to spectral characteristics of chromatic dispersion coefficient for conventional SMF (ITU-T Rec. G.652): here, zero dispersion wavelength of both the fundamental and higher-order guided modes corresponds to wavelength range λ = 1300…1350 nm.

FBG was written in the sample of fabricated FMF 11/125 segment and test series were performed to research a few-mode effects, occurring during laser-excited optical signal propagation over FMF with written FBG, both unperturbed and under the temperature or mechanical actions. Main and periphery peaks were localized on the spectral response of unperturbed FMF FBG, while under the stress besides the expected Bragg wavelength shifting, spectral response smoothed and periphery peaks dropped out. Results of performed theoretical and experimental researches showed a good potentiality for utilization of designed and fabricated twisted FMF 11/125 in various applications of selected order guided mode management as well as in fiber optic sensors.

## Figures and Tables

**Figure 1 sensors-22-03124-f001:**
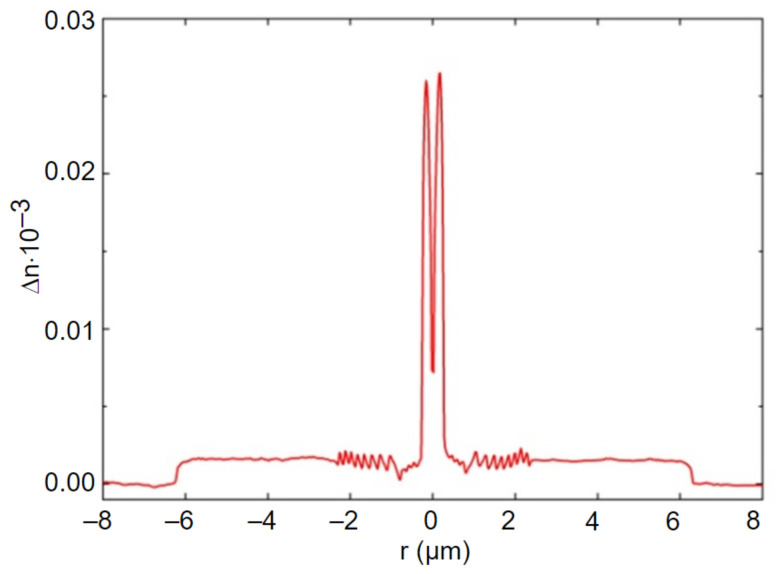
Refractive index profile of pilot FMF preform (measured by refractometer Photon Kinetics P101).

**Figure 2 sensors-22-03124-f002:**
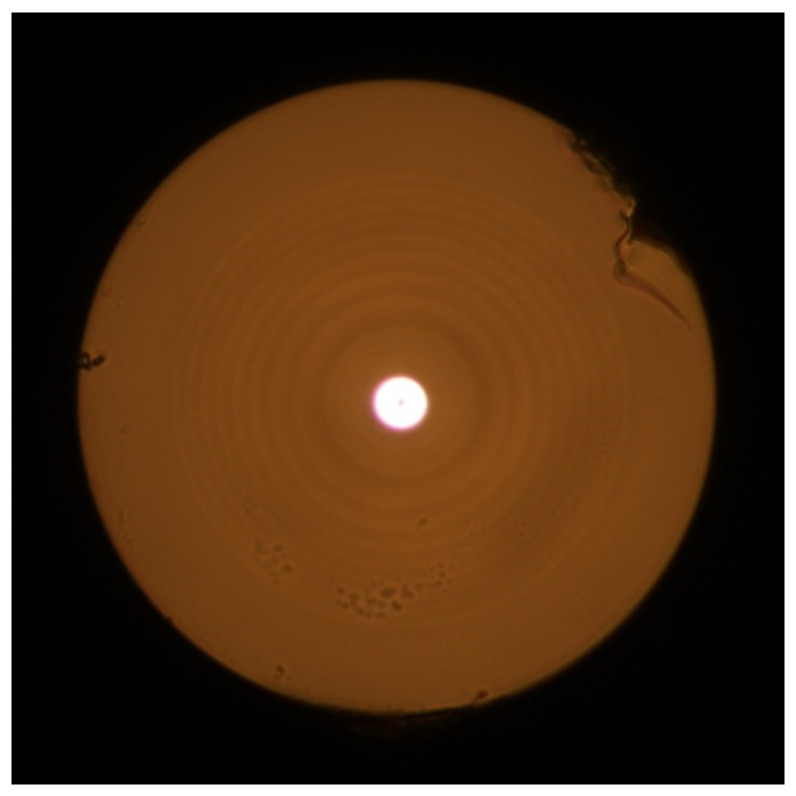
Image of the end-face of fabricated pilot 4-mode FMF 11/125 with numerical aperture *NA* = 0.22 (high-resolution optical microscope Nikon Eclipse N-U).

**Figure 3 sensors-22-03124-f003:**
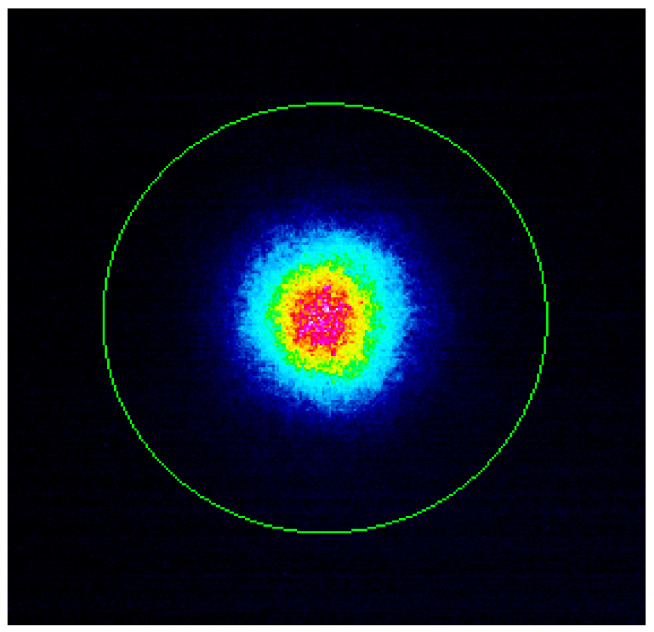
Near field laser beam profile (operating wavelength λ = 1550 nm), measured after propagation over pilot sample of FMF 11/125 by CCD camera DataRay WinCamD-LCM-C-TE.

**Figure 4 sensors-22-03124-f004:**
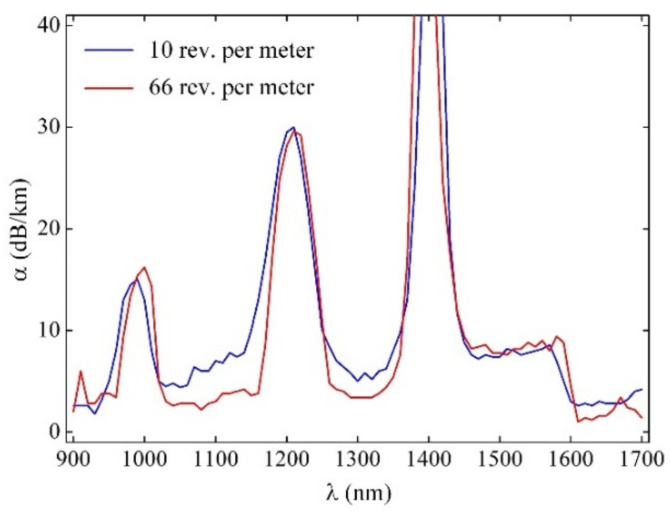
Attenuation of manufactured 50 m length FMF 11/125 samples with induced twisting 10 and 66 rpm.

**Figure 5 sensors-22-03124-f005:**
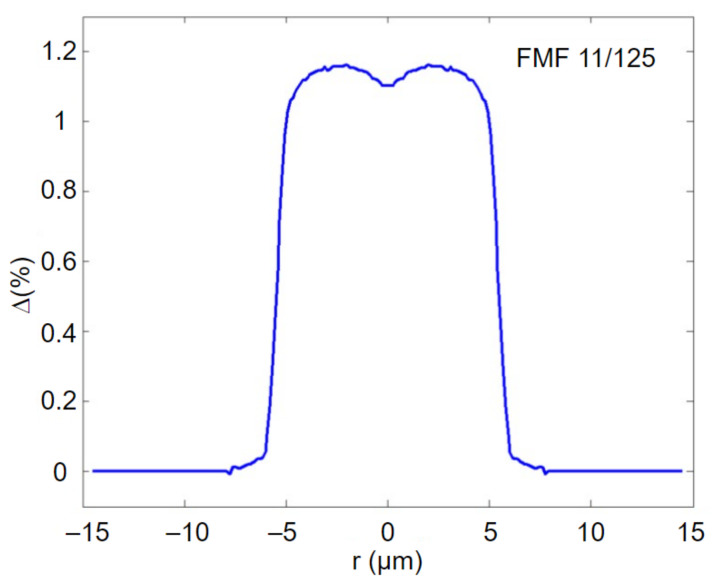
Equivalent quasi-step refractive index profile with improved height, restored by measurement report data.

**Figure 6 sensors-22-03124-f006:**
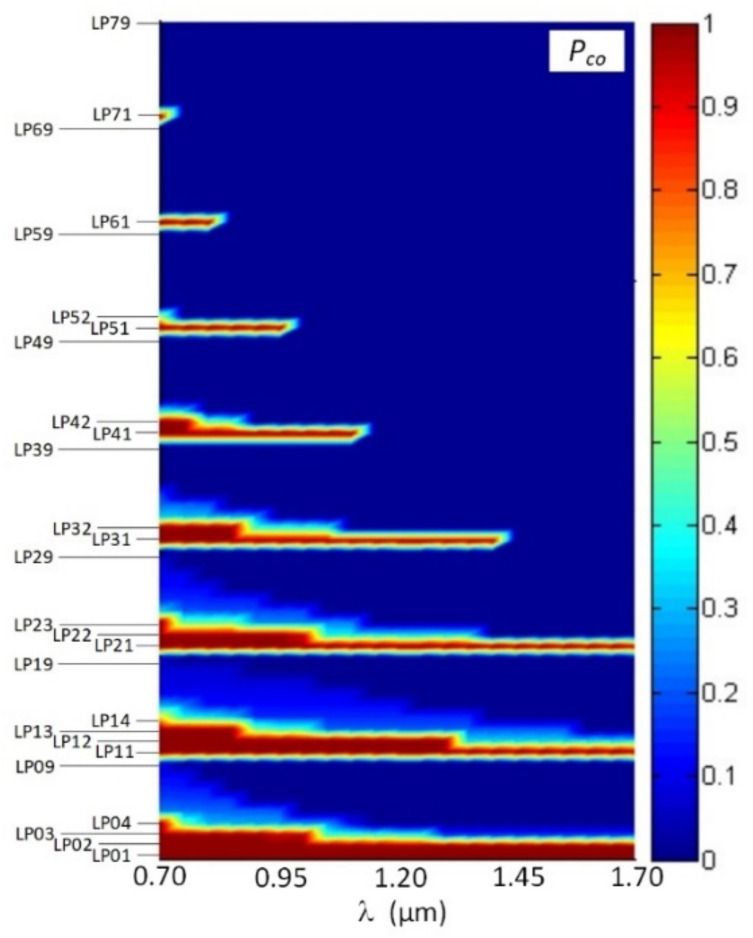
Diagram of the optical confinement factor distribution between modes of FMF 11/125 over the wavelength range λ = 700…1700 nm.

**Figure 7 sensors-22-03124-f007:**
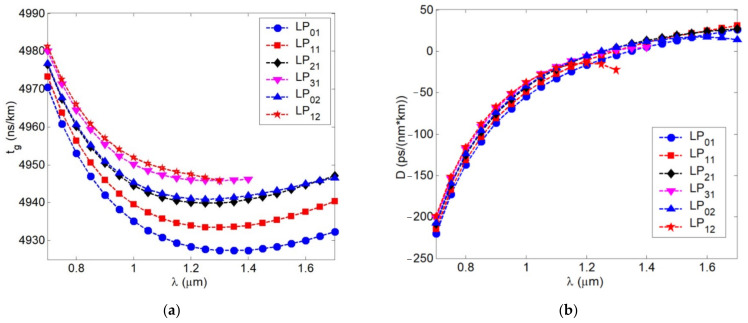
Spectral characteristics of guided mode dispersion parameters: (**a**) mode delay; (**b**) chromatic dispersion coefficient.

**Figure 8 sensors-22-03124-f008:**
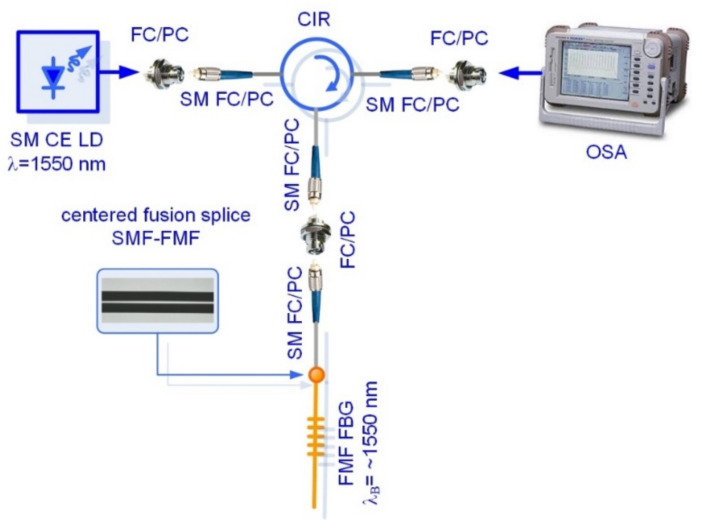
Conventional setup for reflected FBG spectral response measurement: testing of FBG, written on FMF under laser-based few-mode operation.

**Figure 9 sensors-22-03124-f009:**
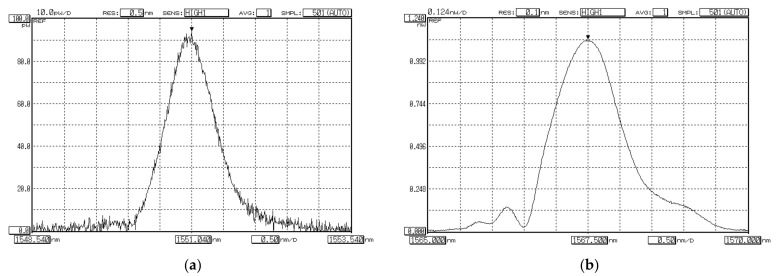
Spectral responses of FBG, excited by laser-source (CE LD, λ = 1550 nm): (**a**) FBG on SMF; (**b**) FBG on FMF 11/125.

**Figure 10 sensors-22-03124-f010:**
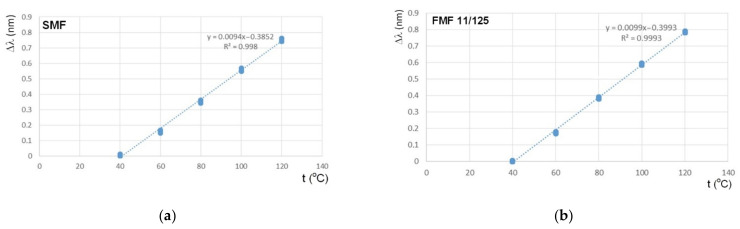
FBG Bragg wavelength λ_B_ shifting sensitivity to the temperature action: (**a**) FBG on SMF; (**b**) FBG on FMF 11/125.

**Figure 11 sensors-22-03124-f011:**
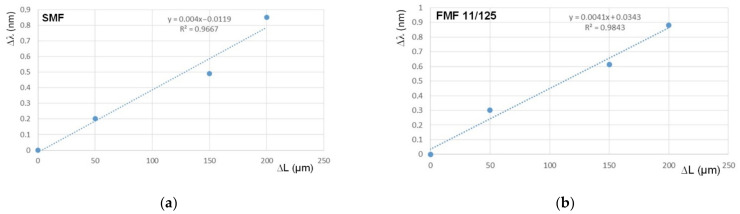
FBG Bragg wavelength λ_B_ shifting sensitivity to the mechanical action: (**a**) FBG on SMF; (**b**) FBG on FMF 11/125.

**Figure 12 sensors-22-03124-f012:**
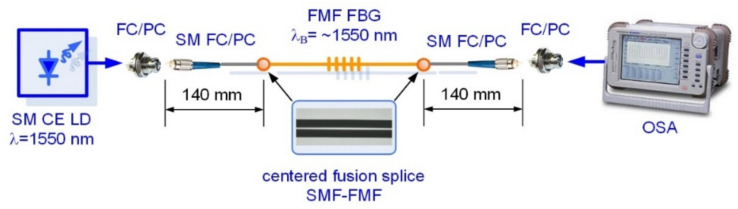
Conventional setup for direct FBG spectral response measurement: testing of FBG, written on FMF under laser-based few-mode operation.

**Figure 13 sensors-22-03124-f013:**
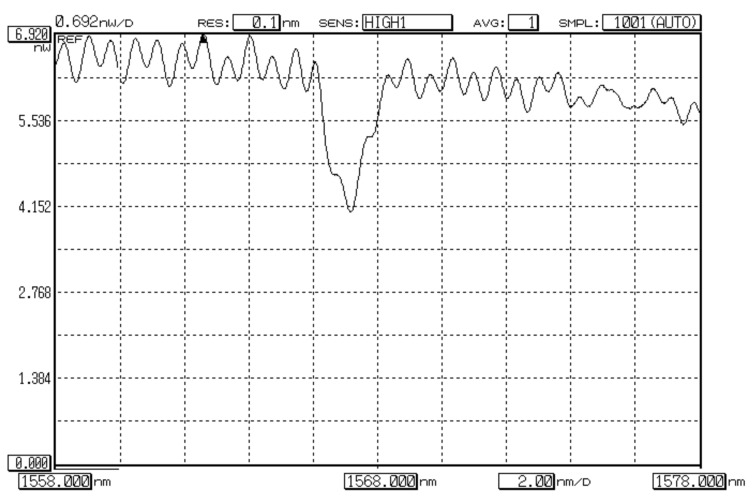
Reference spectral response of unperturbed FBG, written on FMF 11/125.

**Figure 14 sensors-22-03124-f014:**
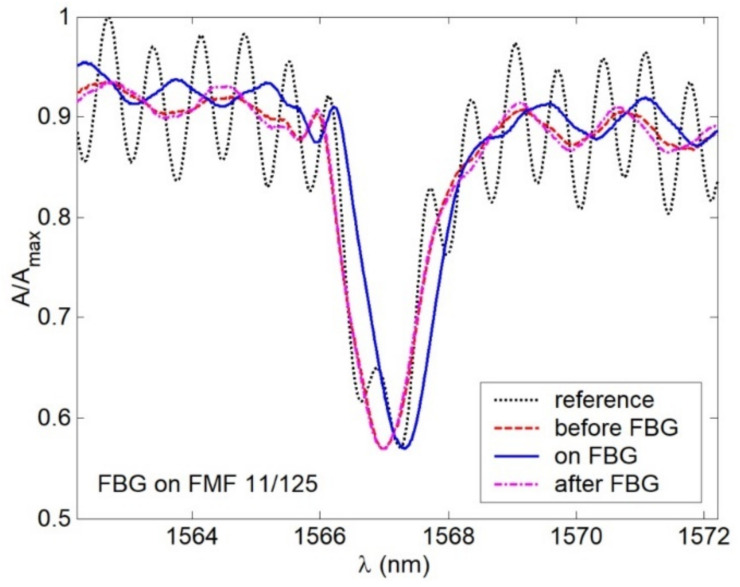
Spectral responses under 15 mm loop before, after and on the FBG, written on FMF 11/125.

**Figure 15 sensors-22-03124-f015:**
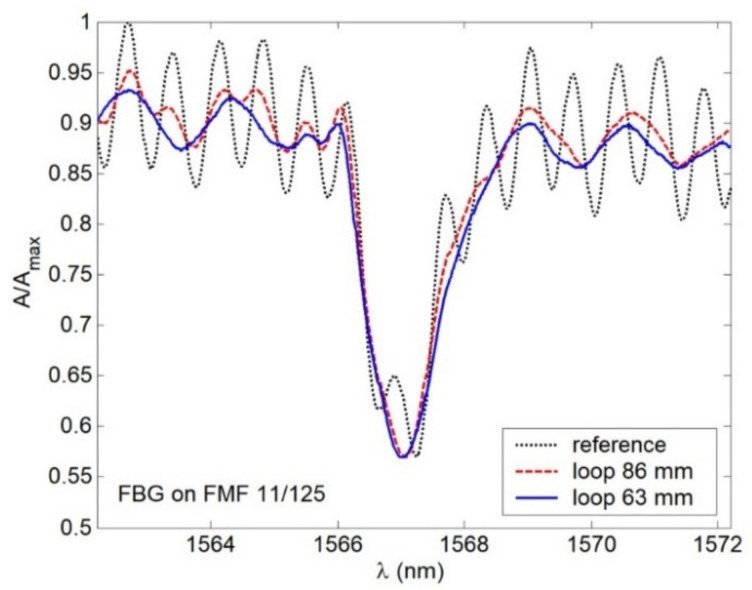
Spectral responses under 86 and 63 mm loops, placed over FMF 11/125 with written FBG.

**Table 1 sensors-22-03124-t001:** Results of optical fiber analysis, performed by rigorous numerical method: step-index optical fibers under various combinations of core diameter and numerical aperture (λ = 1550 nm).

№	Core Diameter, μm	Cladding Diameter, μm	Numerical Aperture *NA*	Mode Composition	*n* _eff_	Δ*n*_eff_
1	8.3	125	0.14	*LP* _01_	1.460478	–
2	8.3	125	0.16	*LP* _01_ *LP* _11_	1.462210 1.457688	0.004522
3	8.3	125	0.18	*LP* _01_ *LP* _11_	1.464263 1.459082	0.005181
4	8.3	125	0.20	*LP* _01_ *LP* _11_	1.466624 1.460940	0.005684
5	8.3	125	0.22	*LP* _01_ *LP* _11_	1.469284 1.463199	0.006085
6	8.3	125	0.24	*LP* _01_ *LP* _11_ *LP* _21_ *LP* _02_	1.472237 1.465821 1.458136 1.457159	0.006416 0.014101 0.015078
7	10	125	0.14	*LP* _01_ *LP* _11_	1.461181 1.457847	0.003334
8	10	125	0.16	*LP* _01_ *LP* _11_	1.463027 1.459219	0.003808
9	10	125	0.18	*LP* _01_ *LP* _11_	1.465177 1.461012	0.004165
10	10	125	0.20	*LP* _01_ *LP* _11_ *LP* _21_ *LP* _02_	1.467622 1.463176 1.457854 1.457139	0.004446 0.009768 0.010483
11	10	125	0.22	*LP* _01_ *LP* _11_ *LP* _21_ *LP* _02_	1.470355 1.465682 1.459875 1.458498	0.004673 0.010480 0.011857
12	10	125	0.24	*LP* _01_ *LP* _11_ *LP* _21_ *LP* _02_	1.473371 1.468510 1.462343 1.460621	0.004861 0.011028 0.012750
13	11	125	0.14	*LP* _01_ *LP* _11_	1.461499 1.458455	0.003044
14	11	125	0.16	*LP* _01_ *LP* _11_	1.463387 1.459994	0.003393
15	11	125	0.18	*LP* _01_ *LP* _11_ *LP* _21_ *LP* _02_	1.465572 1.461914 1.457563 1.457061	0.003658 0.008009 0.008511
16	11	125	0.20	*LP* _01_ *LP* _11_ *LP* _21_ *LP* _02_	1.468048 1.464179 1.459378 1.458241	0.003869 0.008670 0.009807
17	11	125	0.22	*LP* _01_ *LP* _11_ *LP* _21_ *LP* _02_	1.470808 1.466767 1.461642 1.460198	0.004041 0.009166 0.010610
18	11	125	0.24	*LP* _01_ *LP* _11_ *LP* _21_ *LP* _02_ *LP* _31_	1.473847 1.469664 1.464288 1.462650 1.458078	0.004183 0.009559 0.011197 0.015769

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
