# Peer review of "Twisted Few-Mode Optical Fiber with Improved Height of Quasi-Step Refractive Index Profile"

_sensors, 2022, doi:10.3390/s22093124_

Round 1

Reviewer 1 Report

This paper discusses the theoretical conception and fabrication of a chiral FMF 11/125 that provides a 4-mode operation regime over the C-band. An improvement in the height of the step refractive index profile was achieved. The proposed optical fiber presents a high attenuation coefficient, which the authors attribute to the simplified fabrication process of the FMF's preform. The FMF 11/125 can be used for selected order guided mode management and optical fiber sensing.

Although I find the paper's subject very interesting, the english writing was extremely tiring due to some grammatical errors (specially concerning prepositions) and the absence of periods. For this reason, I think this paper needs extensive writing revisions before being ready for publication. 

By far, the "Abstract" and "Introduction" are the sections which deserve more attention. Despite the rest of the paper being better written, it still has several grammatical errors. I suggest that the authors perform a major analysis along with proficient english speakers, and resubmit the paper. 

Author Response

Many thanks for reviewing the paper and for your valuable comments. Based on your comments, the paper is revised: we rewrote and corrected the most part of the manuscript to improve English as well as to eliminate gramma errors

Reviewer 2 Report

Thank the authors for presenting a comprehensive study of a few-mode fibre with enhanced index contrast, including the design, numerical modelling, fabrication and experimental measurement in practical scenarios. However, I think the study is still somehow less convincing before publication in Sensors. A couple of issues that can be improved/tackled include:

  1. For the design of this few-mode fiber, the study concludes with the optimal parameter set (11-micron core diameter, 125-micron cladding diameter and NA 0.22) by comparing with a limited group of other parameter sets as listed in Table 1. The study loses some generality here. why is, for example, the core diameter optmised with only the three values 8.3, 10 and 11 microns?)
  2. A fiber of circular symmetry should not in theory have any difference under twist, as is also the case discussed in the manuscript (Fig. 2) [1]. Surely the fabrication deformation always introduces some chirality by twist, which is however not so convincing to utilize a twist in a circular-core fiber.
  3. The presentation of the study needs improvement. In addition to typos, some important terminations need a proper definition. For example, what is the optical confinement factor? 

[1] X. Xi et. al, Phys. Rev. Lett. 110, 143903.

Author Response

Many thanks for reviewing the paper and for your valuable comments. Based on your comments, the paper is revised, and the revised parts are marked by yellow in the revised paper.

  1. For the design of this few-mode fiber, the study concludes with the optimal parameter set (11-micron core diameter, 125-micron cladding diameter and NA 0.22) by comparing with a limited group of other parameter sets as listed in Table 1. The study loses some generality here. why is, for example, the core diameter optimized with only the three values 8.3, 10 and 11 microns?)

Thank you for your question. We start from 8.3 um due to it is typical core diameter for standard telecommunication single mode optical fibers (ITU-T Rec. G.652). Here we supposed, that even the weak improvement of refractive index profile height may provide desired few-mode regime. However, results of computation shew, that increasing of both core diameter and numerical aperture is required.  

We rewrote corresponding text block in Section 2.

  1. A fiber of circular symmetry should not in theory have any difference under twist, as is also the case discussed in the manuscript (Fig. 2) [1]. Surely the fabrication deformation always introduces some chirality by twist, which is however not so convincing to utilize a twist in a circular-core fiber.

Many thanks for your comment. Please note, that manuscript [1] considered features of twisted singlemode optical fibers (SMF) from the point of view of polarization properties, while few-mode and laser-excited multimode optical fibers (FMFs and MMFs) add new dimension in the space of parameters – limited number of guided modes with specified order. Also, there are known published works, which declare twisted optical fibers as new alternative approach in fiber optic sensing technique, as well as fiber Bragg grating, written in FMFs and MMFs, are demonstrated new effects / additional information in spectral responses under various temperature or mechanical influences. Therefore, we tried to combine mentioned above features in one successfully fabricated and presented in this work new fiber optic element – laser-excited twisted FMF with recorded FBG. So, it differs by:

  • chirality
  • few-mode regime / operation (4 guided modes over “C”-band)
  • recorded FBG

We add necessary comments as well as corresponding reference to the Introduction.

  1. The presentation of the study needs improvement. In addition to typos, some important terminations need a proper definition. For example, what is the optical confinement factor? 

Thank you for your valuable comment. First of all, we rewrote and corrected the most part of the manuscript to improve English as well as to eliminate gramma errors.

Please find definition of the optical confinement factor here:

https://resources.system-analysis.cadence.com/blog/msa2021-a-high-optical-waveguide-confinement-factor-indicates-low-optical-losses

It is also known as normalized mode power, transferred over (concentrated in) the optical fiber core: mode power in core / total mode power

We add comment to corresponding text block in Section 2.

Reviewer 3 Report

A chiral twisted few-mode fiber (FMF) is proposed in this paper. Through simulation and numerical analysis, the parameters of the fiber are obtained, and its refractive index, attenuation, and other properties are analyzed. Finally, for the fabricated FMF, the mode composition, attenuation, and dispersion parameters of the direct spectral response of the optical signal excited by the coherent optical radiation source after writing the FBG are analyzed and compared with the traditional SMF. Here are some suggestions:
1. The theory of this paper is very detailed. In contrast, the part about the fabrication of FMF is relatively insufficient. Authors can add a description of this aspect, such as the device and step diagram of the MCVD method.
2. The content of the “Introduction” part is not enough to explain the importance and necessity of this research of FMF. It is suggested to supplement it.
3. Tables can be added to summarize the performance comparison between FMF and SMF so that readers can more intuitively feel the advantages of FMF.
4. Some font formats in the picture need to be unified.

Author Response

A chiral twisted few-mode fiber (FMF) is proposed in this paper. Through simulation and numerical analysis, the parameters of the fiber are obtained, and its refractive index, attenuation, and other properties are analyzed. Finally, for the fabricated FMF, the mode composition, attenuation, and dispersion parameters of the direct spectral response of the optical signal excited by the coherent optical radiation source after writing the FBG are analyzed and compared with the traditional SMF. Here are some suggestions:

Many thanks for reviewing the paper and for your valuable comments. Based on your comments, the paper is revised, and the revised parts are marked by red in the revised paper

  1. The theory of this paper is very detailed. In contrast, the part about the fabrication of FMF is relatively insufficient. Authors can add a description of this aspect, such as the device and step diagram of the MCVD method.

Thank you for your comment. We prepared FMF preform by regular OptoGear MCVD-station. Here conventional MCVD method was utilized without any customization.

It is well known and described in details in various textbooks and encyclopedias:

https://www.google.com/search?q=modified+chemical+vapor+deposition+(MCVD)+method&newwindow=1&sxsrf=APq-WBvYlsLJLxaU_AIbK4GYHX7nJSKJiw:1647685875269&source=lnms&tbm=bks&sa=X&ved=2ahUKEwjlza-a_NH2AhVhi8MKHXhfBjAQ_AUoAXoECAEQCw&biw=1280&bih=530&dpr=1.5

We fear plagiarism by citing this well-known matter and add just corresponding reference in Section 3.

Also we add the reference on the earlier on published work, concerned with fabrication  of chiral microstructured optical fiber with special geometry: here we described in details performed modification of drawing tower.

  1. The content of the “Introduction” part is not enough to explain the importance and necessity of this research of FMF. It is suggested to supplement it.

Thank you for your comment, we rewrote the introduction.

  1. Tables can be added to summarize the performance comparison between FMF and SMF so that readers can more intuitively feel the advantages of FMF.

Thank you for your comment.

The main objective of work is demonstration of successfully fabricated new complicated fiber optic element, differing by following:

  • chirality
  • few-mode regime / operation (4 guided modes over “C”-band)
  • recorded FBG

We presented results of some tests, but there were no any goals to detect advantages in comparison with SMF due to few-mode operation is qualitatively other than single mode. Therefore, we suppose, direct comparison between SMF and FMF is incorrect. For example, in Section 4 we shew qualitative difference between SMF and FMF FBG responses: there are main and periphery peaks on FMF FBG spectral response, that may provide additional information in comparison with detection of only Bragg wavelength shifting.

  1. Some font formats in the picture need to be unified.

Thank you for your comment, we changed fonts on Fig. 1 from Times New Roman to Arial, unified for other figures.

Round 2

Reviewer 1 Report

The authors have addressed correctly my suggestions. The paper can be accepted in its present form.

Author Response

Thank you

Reviewer 2 Report

Thank the authors for the revision and rewriting of the manuscript. Now I think it is close to publication. However, personally I cannot agree with the authors' comment on twisted circular-core fibers. Circular-core fibers (core is on axis) do not feel the twist simply because the rotational symmetric structure remains unchanged under twist, thus the boundary conditions are exactly the same as the untwisted case if you want to solve the wave equations. It is irrelevant to whether the fiber supports a few modes or just a single one. Surely, as the authors commented, writing Bragg gratings in the core is a way to break the translational symmetry, and it will make twist count if the gratings properly form. Therefore, I believe the manuscript should refer a bit to how the grating is written and that the grating makes the circular-core fiber present chirality.

Author Response

Thank the authors for the revision and rewriting of the manuscript. Now I think it is close to publication. However, personally I cannot agree with the authors' comment on twisted circular-core fibers. Circular-core fibers (core is on axis) do not feel the twist simply because the rotational symmetric structure remains unchanged under twist, thus the boundary conditions are exactly the same as the untwisted case if you want to solve the wave equations. It is irrelevant to whether the fiber supports a few modes or just a single one. Surely, as the authors commented, writing Bragg gratings in the core is a way to break the translational symmetry, and it will make twist count if the gratings properly form. Therefore, I believe the manuscript should refer a bit to how the grating is written and that the grating makes the circular-core fiber present chirality.

Many thanks for your valuable comment.

The work [X. Xi et. al, Phys. Rev. Lett. 110, 143903] declares, that ideal solid circular core axially symmetric optical fiber (with core on axis) is insensitive to twisting from the point of view retention of translation symmetry. Therefore, the boundary conditions are exactly the same as the untwisted case by meaning solution of the wave equation.

However, on practice, real fabricated optical fibers (as well as prepared preforms) are not ideal. They differ by asymmetrical elliptical core and core/cladding eccentricity. It might be supposed, that described above non-ideality of optical fiber may impact on some changes in translation symmetry.

Without going into detail of numerical analysis, earlier on published works confirmed and demonstrated, that twisting may lead to some new specific effects. For example, twisted singlemode optical fibers were presented as fibers with reduced polarization mode dispersion (PMD) [1, 2], while twisting of multimode optical fibers was declared as the method for differential mode delay (DMD) decreasing with total bandwidth improvement [3, 4]. Other papers [5, 6] announced a good potentiality for various applications of twisted optical fibers in fiber optic sensors.

Therefore, we decided to combine several features in one fiber optic element – laser-excited twisted silica few-mode optical fiber (FMF) with recorded FBG. So, it differs by:

  • twisting (twisted FMF)
  • few-mode regime / operation (4 guided modes over “C”-band)
  • FBG, recorded in twisted FMF (we utilized conventional Lloyd interferometric setup workstation)

The main purpose of the work is presentation of designed and successfully fabricated described above fiber optic structure with pilot (preliminary) experimental approbation in measurements of spectral responses to confirm its operability to pass the optical signal and to modify it. Of course, detailed analysis how twisting of FMF as well as, for example, FBG placement in twisted FMF (e.g. on or between revolutions) will impact on optical signal transmission process require additional detailly plan researches. It is beyond of the considered work.

P.S. To avoid some misunderstood, we exchange term “chiral” on “twisted” throughout all the manuscript.

  1. A.C. Jr., Huff, R.G., Walker, K.L. Method of making a fiber having low polarization mode dispersion due to a permanent spin. U.S. Patent 5298047. 1994.
  2. Li, M.-J., Chen, X., Nolan, D.A. Fiber spinning for reducing polarization mode dispersion in single-mode fibers: theory and applications. Proceedings of SPIE. 2003, 247, 97–110.
  3. DiGiovanni, D.J., Golowich, S.E., Jones, S.L., Reed, W.A. Method of making an improved multimode optical fiber and fiber made by method. Patent U.S. 2001/0019652. 2001.
  4. DiGiovanny, D.J., DiMarcello, F.V., Jiang, X.L., Oulundsen, G.E., Pandit, S.P. Multimode optical fiber with increased bandwidth. Patent U.S. 2004/0228590 A1. 2004.
  5. Kopp, V.I., Churikov, V.M., Singer, J., Neugroschl, D. Chiral fiber sensors. Proceedings of SPIE 2010, 7677, 76770U-1‒76770U-6.
  6. Kopp, V.I., Genack, A.Z. Adding twist. Nature Photonics 2011, 5(8), 470–472.